# Biglycan Promotes Cancer Stem Cell Properties, NFκB Signaling and Metastatic Potential in Breast Cancer Cells

**DOI:** 10.3390/cancers14020455

**Published:** 2022-01-17

**Authors:** Kanakaraju Manupati, Ritama Paul, Mingang Hao, Michael Haas, Zhaoqun Christine Bian, Tammy M. Holm, Jun-Lin Guan, Syn Kok Yeo

**Affiliations:** 1Department of Cancer Biology, University of Cincinnati College of Medicine, Cincinnati, OH 45267, USA; kanakami@ucmail.uc.edu (K.M.); ritama.paul@pennmedicine.upenn.edu (R.P.); haomg@ucmail.uc.edu (M.H.); haasmk@ucmail.uc.edu (M.H.); guanjl@ucmail.uc.edu (J.-L.G.); 2Department of Surgery, University of Cincinnati, 231 Albert Sabin Way, Cincinnati, OH 45267, USA; bianzn@ucmail.uc.edu (Z.C.B.); holmty@ucmail.uc.edu (T.M.H.)

**Keywords:** breast cancer stem cells, biglycan, metastasis, NFκB pathway, luminal breast cancer

## Abstract

**Simple Summary:**

Breast cancer stem cells (BCSCs) are a small sub-population of cells within tumors with high metastatic potential. We identified biglycan (BGN) as a prospective molecular target in BCSCs that regulates the aggressive phenotypes of these cells. These findings establish a foundation for the development of therapeutics against BGN to eliminate BCSCs and prevent metastatic breast cancer.

**Abstract:**

It is a major challenge to treat metastasis due to the presence of heterogenous BCSCs. Therefore, it is important to identify new molecular targets and their underlying molecular mechanisms in various BCSCs to improve treatment of breast cancer metastasis. Here, we performed RNA sequencing on two distinct co-existing BCSC populations, ALDH^+^ and CD29^hi^ CD61^+^ from PyMT mammary tumor cells and detected upregulation of biglycan (BGN) in these BCSCs. Genetic depletion of BGN reduced BCSC proportions and tumorsphere formation. Furthermore, BCSC associated aggressive traits such as migration and invasion were significantly reduced by depletion of BGN. Glycolytic and mitochondrial metabolic assays also revealed that BCSCs exhibited decreased metabolism upon loss of BGN. BCSCs showed decreased activation of the NFκB transcription factor, p65, and phospho-IκB levels upon BGN ablation, indicating regulation of NFκB pathway by BGN. To further support our data, we also characterized CD24^−^/CD44^+^ BCSCs from human luminal MCF-7 breast cancer cells. These CD24^−^/CD44^+^ BCSCs similarly exhibited reduced tumorigenic phenotypes, metabolism and attenuation of NFκB pathway after knockdown of BGN. Finally, loss of BGN in ALDH^+^ and CD29^hi^ CD61^+^ BCSCs showed decreased metastatic potential, suggesting BGN serves as an important therapeutic target in BCSCs for treating metastasis of breast cancer.

## 1. Introduction

Breast cancer is a leading cause of mortality in women, and the prevalence of heterogeneity and molecular complexity in these tumors are impediments to effective treatment of the disease [1]. Breast cancer can be classified into various subtypes based on molecular profiling into luminal, (type A and type B), HER-2 overexpressing, normal-like and basal-like (TNBC) [2]. Among these, luminal-type breast cancers account for about 70% of all breast cancers [3] and are associated with relatively better prognosis [4]. However, the development of resistance to endocrine therapy and subsequent disease relapse is still a major problem [5]. Additionally, luminal breast cancer patients also tend to develop tumor recurrence within 5 years of surgery or chemotherapy [6,7]. Therefore, it is necessary to account for heterogenous populations within the tumor to curb resistance when developing targeted therapies, in order to improve the prognosis and survival of these patients.

BCSCs are a subpopulation of cells with distinct tumorigenic properties within a mammary tumor and they drive distant metastasis to the secondary organs [8]. BCSCs have been isolated by using specific cell surface markers, such as CD24, CD44, CD133, CD29, CD61 and ALDH1 from heterogeneous populations within tumors [9]. It has been shown that conventional therapies, such as radiotherapy and chemotherapy, are ineffective in eradicating BCSCs, largely due to their ability to evade apoptosis [10]. Previous studies have also reported that epithelial BCSCs were enriched in central regions of the tumor, whereas mesenchymal BCSCs were found at the invasive front, indicating that distinct BCSCs can co-exist and occupy different niches within a tumor [11]. These observations elevate the risk that diverse BCSC populations may have differential sensitivities to BCSC-targeted therapeutics [12]. BCSCs are predicted to persist in tumors as residual populations and seed metastasis [13]. Therefore, development of specific targeted therapies for various BCSCs to be used in combination is a prospective therapeutic strategy for improvement of survival in breast cancer patients with disease relapse.

Biglycan (BGN) is a member of the small leucine rich proteoglycan (SLRP) family and is comprised of side chains of chondroitin sulfate or dermatan sulfate and a 42-kDa core protein [14]. It is integrated within the extracellular matrix (ECM) under physiological environments and also found to be expressed on the cell surface [15]. BGN plays a major role in various cellular processes including cell migration, adhesion, inflammation, cell growth, regulation of autophagy, apoptosis and modulation of matrix assembly [16]. Higher BGN expression was observed in tumor tissues relative to adjacent normal tissues for multiple cancer types [17]. Furthermore, BGN has been implicated in various tissue specific oncogenesis such as pancreatic, gastric, endometrial, colon and bladder cancer [15]. BGN binds to TLR4 and activates the NF-κB pathway in colon cancer cells, leading to epigenetic silencing of immunosuppressive siglec-7 ligand glycans [18]. High expression of BGN in gastric cancer tissues was associated with lymph node metastasis by activation of the FAK signaling pathway [15]. These findings indicate that BGN plays a role in tumorigenesis and metastasis. However, the underlying role of BGN in breast cancer and BCSCs remains elusive.

In this study, we focused on identification of a prospective therapeutic target in two different BCSC populations (ALDH^+^ and CD29^hi^ CD61^+^) sorted from luminal subtype cancer cells (PyMT-driven) using RNA sequencing. Further, the identified molecular target, BGN, was investigated for its role in tumorigenic phenotypes such as BCSC migration, invasion and tumorsphere formation. Similar results were observed with CD24^−^/CD44^+^ BCSCs sorted from a human luminal breast cancer cell line, MCF-7. We also revealed that BGN is important for metabolism of BCSCs confirmed by glycolytic and mitochondrial stress tests. Furthermore, BGN depletion in BCSCs attenuated the NFκB pathway by downregulating phosphorylation of p65 and IκB subunits. Finally, ALDH^+^ and CD29^hi^ CD61^+^ BCSCs depleted of BGN exhibited reduced metastasis, suggesting BGN as a potential therapeutic target in BCSCs to inhibit breast cancer metastasis.

## 2. Results

### 2.1. Transcriptomic Analysis Reveals Enrichment of Extracellular Matrix Genes and BGN Expression in PyMT-Driven Tumor BCSCs

In our previous study, we demonstrated that MMTV-PyMT mammary tumors contained two distinct populations of BCSCs demarcated by CD29^hi^ CD61^+^ and ALDH^+^ respectively [19,20]. These distinct BCSCs exhibit different properties and have distinct dependencies on signaling pathways. To gain further insights into the characteristics of these BCSCs, we isolated these populations by flow cytometry using dissociated cells from MMTV-PyMT tumors. Tumor cells were sorted into bulk cancer cells (P0), CD29^hi^ CD61^+^ BCSCs (P1) and ALDH^+^ BCSCs (P2). RNA-sequencing and transcriptomic analysis was then performed on the three sorted populations (P0, P1 and P2 in duplicates). A total of 153 genes were found to be differentially regulated in P1 cells relative to P0 bulk tumor cells and 106 genes differentially expressed in P2 cells relative to P0 (Figure 1A). Among the BCSC populations, 27 genes were common to both comparisons (Figure 1A). Interestingly, when the differentially expressed genes from both comparisons were analyzed for enriched KEGG pathways, “ECM Receptor interaction” and “Focal adhesion” were among the top four categories present in both sets of comparisons (Figure 1B). The enrichment of extracellular matrix (ECM) gene expression in the respective BCSC populations is visualized through a heat map (Figure 1C). Next, we validated the differentially regulated genes using qRT-PCR. Basal markers such as *Krt5*, *Trp63*, *Lama3*, *Ccl2*, along with ECM genes *Postn*, *Col7a1*, *Col2a1* and *Col17a1* were significantly upregulated in CD29^hi^ CD61^+^ BCSCs (P1) (Figure 1D). Contrastingly, *Aldh1a3*, *Ccl5*, *Ccl11* and *Ccl7* expression were significantly higher in ALDH^+^ BCSCs (P2) as compared to bulk cancer cells (P0), indicating that our sorted populations were indeed pure and RNA sequencing was reflecting the expected characteristics of these BCSC populations (Figure 1E). Consistent with enrichment of ECM genes in BCSC populations (Figure 1B), the expression of biglycan (*Bgn*), *Dcn*, *Col6a1*, *Col3a1* and *Col1a2* were upregulated in both BCSC populations (Figure 1F). Among these, BGN was identified as an important common molecular target in BCSCs to be investigated further, considering its role and function in CSCs have not been described. Thus, BGN expression was evaluated at protein level in bulk cancer cells and BCSCs using immunoblotting. These results showed that BGN protein expression levels were markedly higher in ALDH^+^ BCSCs (Figure 1G) and CD29^hi^ CD61^+^ BCSCs (Figure 1H) as compared to bulk cells. To further validate our results from PyMT tumors, we also utilized a human luminal breast cancer cell line, MCF-7, and its sorted BCSCs, CD24^−^/CD44^+^ for determining BGN expression. Concordantly, high BGN expression was detected in MCF-7 derived CD24^−^/CD44^+^ BCSCs as compared to their respective bulk cancer cells (Appendix A). Overall, the transcriptomic analyses revealed the upregulation of ECM associated genes in distinct BCSC populations and BGN as a prospective common molecular target in both BCSC populations.

### 2.2. Genetic Depletion of BGN Leads to Reduced Expression of BCSC Markers and Tumorsphere Forming Ability

To elucidate the role of BGN in BCSCs, we generated CRISPR mediated knockout of BGN in FF99 cells (PyMT tumor derived cells, [21]) using mouse specific sgRNA. Next, we sorted CD29^hi^ CD61^+^ and ALDH^+^ CSCs from control sgRNA FF99 and BGN sgRNA FF99 cells using FACS based on surface markers, CD29, CD61 and ALDH enzyme activity as described previously [19]. BGN protein expression was attenuated with BGN sgRNA as compared to control sgRNA in these cells (Figure 2A) and their sorted BCSC subpopulations, ALDH^+^ (Figure 2B,C) and CD29^hi^ CD61^+^ (Figure 2D,E) as confirmed by immunoblotting and immunofluorescence. Significant reductions in the number of ALDH^+^ (Figure 2F,G) and CD29^hi^ CD61^+^ (Figure 2H,I) BCSCs were observed in BGN sgRNA FF99 cells relative to control sgRNA FF99 cells using FACS analysis. Furthermore, when subjected to tumorsphere formation assays, both CD29^hi^ CD61^+^ and ALDH^+^ BCSCs formed smaller sized spheres (Figure 2J,L) upon BGN depletion and quantification revealed significantly decreased numbers of spheres as compared to control sgRNA (Figure 2K,M). In the same way, MCF-7 cells were utilized for isolation of BCSCs using the markers CD24^−^/CD44^+^, after shRNA mediated knockdown of BGN using human BGN specific shRNA and control shRNA (Appendix A). Similarly, the number of CD24^−^/CD44^+^ and ALDH^+^ BCSCs were significantly reduced in BGN shRNA expressing MCF-7 cells compared to control shRNA expressing cells (Appendix A). CD24^−^/CD44^+^ BCSCs formed significantly fewer tumorspheres with knockdown of BGN compared to control shRNA (Appendix A), indicating BGN is important for self-renewal of BCSCs. Altogether, these results suggested that BGN plays a role in maintaining BCSC properties in breast cancer.

### 2.3. Depletion of BGN Reduces the Motility and Invasion of BCSCs

BCSCs have an increased propensity to invade and migrate from the primary tumor site to secondary sites such as lung, liver, bone and brain for the formation of metastases [22]. Thus, we evaluated the migration and invasion properties of CD29^hi^ CD61^+^ and ALDH^+^ BCSCs, using wound healing and matrigel invasion assays. Depletion of BGN in CD29^hi^ CD61^+^ (Figure 3A,B) and ALDH^+^ (Figure 3C,D) BCSCs resulted in a significant decrease in the number of migrated cells compared with control sgRNA expressing cells. Similarly, shRNA mediated knockdown of *BGN* in MCF-7 derived CD24^−^/CD44^+^ BCSCs also showed a significantly reduced number of migrated cells compared to control shRNA cells (Appendix A), suggesting a positive role of BGN in regulation of BCSC migration. Further, a significant reduction in the number of invaded cells was also observed in BGN sgRNA expressing CD29^hi^ CD61^+^ (Figure 3E,F) and ALDH^+^ (Figure 3G,H) BCSCs than control sgRNA expressing cells, indicating an important role of BGN in the regulation of invasive capacity of BCSCs. Likewise, BGN shRNA mediated knockdown in CD24^−^/CD44^+^ BCSCs also displayed a significantly lower number of invaded cells than control shRNA cells (Appendix A). Altogether, these results demonstrated that BGN plays a role in promoting migration and invasion capacity of BCSCs.

### 2.4. Bioenergetic Role of BGN in BCSCs

In order to determine whether BGN is also important for BCSC metabolism, we performed glycolytic and mitochondrial flux assays. The glycolysis stress test is depicted by a kinetic graph plotting ECAR (extracellular acidification rate) before and after consecutive injection of modulators (glucose, oligomycin and 2-deoxyglucose) of glycolysis (Figure 4A). CD29^hi^ CD61^+^ and ALDH^+^ BCSCs from FF99 cells with or without BGN depletion were compared. The rate of basal glycolysis was significantly lower in BGN sgRNA expressing CD29^hi^ CD61^+^ and ALDH^+^ BCSCs (Figure 4B) compared with control sgRNA expressing cells. The induction of extracellular acidification with oligomycin was significantly reduced in BGN sgRNA expressing CD29^hi^ CD61^+^ and ALDH^+^ BCSCs (Figure 4C) compared to control cells, indicating decreased glycolytic capacity in BGN deficient BCSCs. We also observed a significantly lower glycolytic reserve in BGN sgRNA expressing CD29^hi^ CD61^+^ and ALDH^+^ (Figure 4D) BCSCs compared with control sgRNA expressing cells. Similarly, CD24^−^/CD44^+^ BCSCs from MCF7 cells showed a reduction in basal glycolysis (Appendix A), glycolytic capacity (Appendix A) and glycolytic reserve (Appendix A) with knockdown of BGN. This data indicates that BGN plays a major role in regulation of glycolysis in BCSCs.

Apart from glycolysis, we also examined the effect of BGN depletion on mitochondrial respiration in BCSCs by performing mitochondrial stress tests. The oxygen consumption rate (OCR) was measured in BCSCs with or without BGN using sequential injection of the mitochondrial inhibitors oligomycin, FCCP and combined rotenone and antimycin A (Figure 4E). BGN sgRNA expressing CD29^hi^ CD61^+^ and ALDH^+^ BCSCs showed a significantly lower rate of basal mitochondrial respiration compared to control sgRNA expressing cells (Figure 4F). The FCCP induced maximal mitochondrial respiration rate was also significantly decreased in BGN sgRNA expressing CD29^hi^ CD61^+^ and ALDH^+^ BCSCs (Figure 4G). Moreover, BGN depletion in CD29^hi^ CD61^+^ and ALDH^+^ BCSCs, exhibited significantly lower spare respiratory capacity (Figure 4H) and ATP production (Figure 4I) than control BCSCs., Additionally, the proton leak was markedly reduced in both BGN deleted CD29^hi^ CD61^+^ and ALDH^+^ BCSCs compared to their respective control cells (Figure 4J). CD24^−^/CD44^+^ BCSCs from MCF-7 cells also had a significant reduction in basal mitochondrial respiration (Appendix A), maximal mitochondrial respiration (Appendix A), spare respiratory capacity (Appendix A), ATP production (Appendix A) and proton leak (Appendix A) with knockdown of BGN as compared to control cells. Overall, these data demonstrated that BGN depletion in BCSCs decreased glycolytic flux and mitochondrial respiration.

### 2.5. BGN Regulates NFκB Signaling in BCSCs

BGN has been shown to regulate various signaling pathways in tissue specific cancers such as p38 signaling in colon cancer [16], NFκB signaling in gastric cancer [23] and PI3K-Akt-NFκB signaling axis in retinoblastomas [24]. Thus, we performed immunoblotting to identify the signaling pathways that were regulated by BGN in BCSCs. Interestingly, BGN sgRNA expressing CD29^hi^ CD61^+^ BCSCs had a significant reduction of phosho-p65 expression compared to control sgRNA expressing cells (Figure 5A,B). Similarly, phospho-IκBα levels, were significantly decreased in BGN sgRNA expressing CD29^hi^ CD61^+^ BCSCs. In the same way, ALDH^+^ BCSCs expressing BGN sgRNA showed significantly reduced p-p65 and p-IκBα compared to control sgRNA expressing BCSCs (Figure 5C,D). These data indicated that BGN regulates NF-κB signaling in BCSCs and represents a prospective mechanism for the changes in BCSC phenotypes observed upon depletion of BGN.

### 2.6. BGN Promotes BCSC Mediated Metastasis in Breast Cancer

Metastasis occurs due to the high migratory and invasive potential of BCSCs in a tumor [25]. High BGN expression in BCSCs promoted increased tumorigenic phenotypes such as migration and invasion in vitro. Therefore, to evaluate the role of BGN in breast cancer metastasis, we performed experimental metastasis assays. We injected 5 × 10^5^ control sgRNA or BGN depleted CD29^hi^ CD61^+^ or ALDH^+^ BCSCs into the tail veins of nude mice. As expected, mice that received BGN depleted CD29^hi^ CD61^+^ BCSC showed significantly decreased numbers of lung metastatic nodules compared to mice injected with control sgRNA CD29^hi^ CD61^+^ BCSCs (Figure 6A,B). Similarly, mice receiving ALDH^+^ BCSCs with loss of BGN exhibited lower number of metastatic lung nodules compared to control sgRNA ALDH^+^ BCSCs (Figure 6C,D). Furthermore, examination of H- and E-stained lung tissues showed smaller metastatic foci in BGN depleted CD29^hi^ CD61^+^ mice compared to control sgRNA CD29^hi^ CD61^+^ BCSCs (Figure 6E). Similarly, lung tissues from mice injected with BGN depleted ALDH^+^ CSCs showed smaller metastatic foci compared to mice injected with control sgRNA ALDH^+^ BCSCs (Figure 6F), indicating BGN is playing an important role in BCSC mediated breast cancer metastasis.

### 2.7. Expression of BGN in Human Breast Cancers Is Associated with Worse Prognoses

To further explore associations implicating a potential role for BGN in human breast cancers, we interrogated data from TCGA using the Gepia online software. We found that BGN mRNA expression was significantly higher in breast tumors as compared to normal breast tissue samples (Figure 7A). Additionally, clinical data showed that high *BGN* expression was significantly associated with poorer overall survival of breast cancer patients compared to low *BGN* expression (Figure 7B). Furthermore, high *BGN* expression was also associated with reduced disease-free survival of breast cancer patients (Figure 7C). Collectively, these data demonstrate that BGN plays a role in maintaining BCSC properties and is a potential therapeutic target to inhibit breast cancer metastasis.

## 3. Discussion

Breast tumors are comprised of populations of cells that are heterogenous at genetic and morphological levels [26]. These populations have been known to differ in their functions at molecular level and exhibit different phenotypes [27]. Therefore, specific cell surface markers have been used to investigate these populations for identification of new molecular targets to improve therapeutic strategies [8]. Currently, the CSC surface markers CD24, CD44, CD133 and ALDH have been widely used in breast cancer for the isolation of BCSCs [28,29]. A recent study has shown that CD24^−^/CD44^+^ CSCs were associated with cell proliferation and tumor progression, whereas ALDH^+^ CSCs were more associated with metastasis, indicating that these two markers enrich for different CSC populations with diverse phenotypes [30]. In mouse mammary tumors, CD29^hi^CD61^+^ and ALDH^+^ markers have enabled the enrichment of separate CSC populations [31]. These BCSC markers identify distinct subpopulations in mouse mammary tumors and differ in their tumorigenic phenotypes, signaling mechanisms, tumor and metastatic progression [19,20]. In this study, we identified a new molecular target, BGN which was upregulated in both BCSCs (CD29^hi^CD61^+^, ALDH^+^) as compared to bulk cancer cells. Due to the lower expression levels of BGN in bulk cells, BGN KO in bulk cells also showed a significant decrease in sphere formation (Appendix A) and migration (Appendix A), albeit the decrease was less pronounced as compared with BCSCs. This is in line with the observation of higher BGN expression in BCSCs but minimal expression in non-CSCs populations (Figure 1). We also observed residual sphere forming ability in BGN KO cells but we cannot rule out the possibility that this might be due to inefficient BGN depletion in these cells (Figure 2A,B,D). Overall, the silencing of BGN in BCSCs reduced the enrichment of stem cell populations and tumorsphere formation, indicating a crucial role for BGN in maintaining BCSC phenotypes.

BGN has been shown to have functions as a tumor suppressor and an oncogene depending on cellular origin in various tissue specific cancers [32]. BGN inhibition in bladder cancer enhanced proliferation of tumor cells demonstrating that BGN acts as a tumor suppressor [33]. Similarly, BGN overexpression in pancreatic cancer cells inhibited the growth of cancer cells by stimulating cell cycle arrest indicating a growth suppressive function [34]. In contrast, we observed that BGN attenuation in BCSCs decreased tumorigenic phenotypes, migration and invasion suggesting its oncogenic function in breast cancer. Likewise, others also reported that knockdown of BGN in colon cancer cells depicted reduced proliferation, migration, invasion and enhanced apoptosis by upregulating the expression of p21 and p27 [16]. We also found that depletion of BGN in BCSCs only affected the growth curves of these cells at later time points (72 h) (Appendix A). Moreover, we found an increased number of apoptotic cells at 72 h in BGN KO BCSCs as compared to control cells (Appendix A). Thus, we interpret this result as BGN being important for survival of BCSCs that impacts the growth curve of these cells at later timepoints, but BGN does not appear to be regulating proliferation of BCSCs (which should have a prominent effect on growth curves even in the early stages). For the first time, we have demonstrated that BGN regulation on metabolism of BCSCs as depicted by reduced activity of glycolysis and mitochondrial respiration in BGN depleted cells. Recent studies in human prostate cancer showed that upregulation of BGN was associated with tumor progression and poor prognosis, indicating that it serves as a marker to determine the aggressiveness of prostate cancer [35]. Interestingly, we also observed that BGN inhibition in BCSCs showed enhanced chemosensitivity in presence of cisplatin and paclitaxel as compared to control cells (Appendix A) indicating a role of BGN in chemoresistance. BGN regulates various signaling pathways such as the p38 signaling pathway in colon cancer cells and PI3K-Akt-NFκB pathway in retinoblastoma cells [16,24]. Consistent with previous studies showing a role of NFκB signaling in CSCs [36,37,38], we also observed the activation of NFκB pathway in CSCs, which was attenuated with loss of BGN, suggesting that this pathway is playing a major role in the tumorigenic phenotypes of BCSCs and its enrichment and metastasis.

In addition to the role of BGN expressed within cancer cells, it is also available in a secreted soluble form which is involved in various biological functions [39,40]. BGN is produced by macrophages upon stimulation of cytokines such as IL-6, IL-1β, TGF-α and is subsequently deposited in the ECM [41]. The secreted form of BGN from tumor endothelial cells (TECs) promoted tumor angiogenesis and metastasis [42,43]. The soluble BGN has been studied as an important molecular target for anti-inflammation and anti-angiogenic therapy [44]. Autophagy regulation by soluble BGN was also observed in macrophages through CD44-TLR4 signaling in renal ischemia reperfusion injury [45]. However, the role of soluble BGN in breast cancer progression and metastasis is unclear; therefore, future studies are required to delineate the specific contributions of intracellular and soluble BGN. Knockout of BGN in E0771 tumor bearing mice reduced tumor angiogenesis, metastasis to the lung and improved efficacy of chemotherapy [40]. Cancer metastasis was observed with overexpression of BGN in gastric cancer tissues [15]. For the first time, we have demonstrated that regulation of BGN in BCSCs mediated breast cancer metastasis.

## 4. Materials and Methods

### 4.1. Cell Culture and Reagents

MMTV-PyMT tumor derived primary tumor cells (FF99) and BCSCs (ALDH^+^ and CD29^hi^CD61^+^ sub-populations) were cultured in DMEM/F12 growth medium supplemented with 10% FBS, 1% penicillin-streptomycin, 10 ng/mL EGF and 20 mg/mL insulin. Cell lines were routinely tested for mycoplasma contamination. The BCSCs were utilized for the subsequent experiments at P0 following isolation using flow cytometry. Cells were cultured in mammosphere medium for 7 days from an initial density of 1 × 10^4^ cells/well in ultra-low attachment 6-well plates [46] (Corning, 3471). Lipofectamine 3000 reagent (Invitrogen, L3000) was used for transfections of human BGN shRNA (Catalogue number/sequence) and mouse BGN sgRNA.

### 4.2. Isolation of BCSCs Using FACS

Fluorescence-activated cell sorting (FACS) was used for isolation of different BCSCs populations. CD29 and CD61 antibodies were used for isolation of CD29^hi^ CD61^+^ BCSCs from FF99 cells expressing control sgRNA and BGN sgRNA, as described previously [19]. Briefly, cells were incubated with CD29 and CD61 antibodies for 20 min at 4 °C followed by PBS wash and isolated using flow cytometry (FACS Aria, BD Biosciences, San Jose, CA, USA). Similarly, human CD24 and CD44 antibodies were used for isolation of CD24^−^/CD44^+^ BCSCs from MCF7 cells [47] expressing control shRNA and BGN shRNA followed by culture for successive experiments. For isolation of ALDH^+^ BCSCs, cells were suspended in aldefluor buffer and incubated with ALDH reagent (StemCell Technologies, Vancouver, BC, Canada for 40 min. at 37 °C according to manufacturer’s instructions. Flow Jo software was used for data analysis. Antibodies used for flow cytometry include: CD29-FITC (Biolegend, 102206), CD61-PE (Biolegend, 104308), CD24-FITC (Biolegend, 311104, San Diego, CA, USA), CD44-PE (Bioledgend, 338808) and Aldefluor assay kit (Stemcell Technologies, 01700, Vancouver, BC, Canada).

### 4.3. RNA Sequencing

RNA-sequencing was conducted by the Genomics, Epigenomics, and Sequencing Core in University of Cincinnati. In short, RNA isolation was performed from sorted cells and NEB-Next Poly(A) mRNA Magnetic Isolation Module (New England BioLabs, Ipswich, MA, USA) was utilized for targeted RNA enrichment. Consequently, PrepX mRNA Library Kit (WaferGen, Fremont, CA, USA combined with Apollo 324 NGS automated library prep system was employed for preparation of cDNA libraries. Cluster generation was performed using cBot, whereas HiSeq sequencing were carried out using the HiSeq systems (Illumina, San Diego, CA, USA). Fordifferential gene expression analysis, sequence reads were aligned to the genome using standard Illumina sequence analysis pipeline, with assistance from The Laboratory for Statistical Genomics and Systems Biology in the University of Cincinnati. RNA sequencing data were deposited in the GEO database under the accession number GSE189223 [48].

### 4.4. Isolation of RNA and Quantitative RT-PCR

Total RNA was extracted from the cells using RNAeasy kit (Qiagen, 74104, Hilden, Germany) followed by cDNA synthesis with Superscript III first strand synthesis kit (Invitrogen, Waltham, MA, USA). Primers of various genes were mixed with SYBR Green and cDNA for amplification using quantitative real-time PCR (BioRad CFX thermocycler, Hercules, CA, USA). Gene expression analysis was carried out as described previously.

### 4.5. Immunoblotting

As previously described [49], cell lysates were prepared in modified RIPA lysis buffer and then subjected to SDS–PAGE for immunoblotting analysis with the following primary and HRP-conjugated secondary antibodies: BGN antibody (Proteintech 16409-1-AP, Rosemont, IL, USA) P-p65 (Cell Signaling, 3033, Danvers, MA, USA), p65 (Cell Signaling, 8242), pIKB (Cell Signaling Technology, 2859), IkB (Cell Signaling Technology, 9242), Vinculin (Sigma Aldrich, V4505, St. Louis, MO, USA, Anti-mouse IgG, HRP-linked Antibody (Cell Signaling Technology, 7076) and Anti-rabbit IgG, HRP-linked Antibody (Cell Signaling Technology, 7074).

### 4.6. BGN Gene Silencing

For CRISPR knockout, BGN guide RNA oligos were annealed using T4 PNK (NEB M0201S) and ligation buffer at 37 °C for 30 min and 95 °C for 5 min in a thermocycler. Next, ligation of annealed oligos and lentiCRISPRv2 plasmid was carried out using quick ligase enzyme (NEB M2200S) at room temperature for 10 min followed by transformation into Stbl3 bacteria for plasmid amplification. For shRNA knockdown of BGN, we utilized pLV-BGNshRNA-Puro plasmid obtained from Cyagen Biosciences. Further, *HEK-293T* cells were transfected with either *lentiCRISPRv2-BGNsgRNA-Puro* or *lentiCRISPRv2-CtrlsgRNA-Puro* (control) vector (Addgene, Watertown, MA, USA) along with *pMD2.G* and *psPAX2* using Lipofectamine 3000 for the generation of lentiviral particles. FF99 cells were transduced with lentiviral particles using polybrene (5 μg/mL) followed by puromycin selection for the stable cell line generation.

### 4.7. Scratch (Wound Healing) Assay

Cells were plated in 12-well plate at a density of 2 × 10^5^ cells/well. A scratch wound in the monolayer was made in the center of the well with a sterile tip. The floating cells were removed with a PBS wash and culture medium returned. At different time intervals images were captured for analysis of migration rate as described previously [8].

### 4.8. Cell Invasion

Invasion chambers were coated with growth factor reduced Matrigel (Corning, 354230, Corning, NY, USA) for 1 h at 37 °C. Cells at a density of 50,000 cells/well added in the upper chamber with base medium and the lower chamber was filled with medium supplemented with 10% FBS. Cells were incubated for 16 h at 37 °C. Cells remaining on the top of the membrane were removed. Cells that invaded the lower chamber and attached to the plate surface were quantified by fixing with 4% PFA, staining with crystal violet and counting [8].

### 4.9. Immunofluorescence Analysis

Cells were cultured on glass coverslips in 6-well plates for 24 h at 37 °C and were then fixed with 4% PFA for 20 min. After blocking with 1% BSA, cells were incubated with primary antibodies overnight at 4 °C, then stained with fluorochrome-conjugated secondary antibodies. Images were acquired using a confocal microscope (Zeiss LSM710 confocal laser scanning microscope).

### 4.10. Glycolysis and Mitochondrial Stress Test

The XF96 extracellular flux analyzer (Seahorse Bioscience) was used to measure the glycolysis and mitochondrial respiration in BCSCs as described previously [50]. Briefly, cells were seeded in the 96-well culture plate (Seahorse Bioscience) at a density of 20,000 cells/well overnight. For the mitochondrial stress test, cells were incubated with XF base medium (pH 7.4) supplemented with 5.5 mM glucose, 2 mM glutamine and 1mM sodium pyruvate for 1 h at 37 °C in a non-CO_2_ incubator. Oxygen consumption rate (OCR) was then measured by sequential injection of Oligomycin A (Sigma, 75351, St. Louis, MO, USA), FCCP (Carbonyl cyanide 4-(trifluoromethoxy) phenylhydrazone) (Sigma, C2920), and Rotenone plus Antimycin A (Sigma, R8875 and A8674). For the glycolysis stress test, cells were incubated with XF base medium (pH 7.4) supplemented with 2mM glutamine for 1 h at 37 °C in a non-CO_2_ incubator. ECAR (Extracellular acidification rate) was measured by consecutive injection of d-Glucose, Oligomycin A, and 2-deoxy-d-glucose. Assay well cell counts were measured using Cyquant Cell Proliferation Assay (Invitrogen, C7026, Waltham, MA, USA to normalize the OCR and ECAR values in each well. The Seahorse XF cell Mito stress test and glycolysis stress test report generators were used to analyse the data.

### 4.11. Mice Tumor Transplants and Metastatic Model

Animal Studies and biosafety protocols have been performed according to local, state, and federal regulations. For the generation of the metastatic model, cells (5 × 10^5^) were injected through tail-vein in athymic *nude* mice. Mice were monitored for 2 months for the formation of metastatic lung nodules.

### 4.12. Statistical Analysis

Results are depicted as mean ± SEM or standard deviation (SD) of at least three or more individual experiments with biological triplicate samples. Images represent typical experiments reproduced at least thrice with similar results. Statistical tests were performed between test groups and their respective controls, using one/two-way ANOVA followed by Tukey’s post-hoc analysis or student’s paired-*t*-test with *p*-values ≤ 0.05 considered significant.

## 5. Conclusions

In conclusion, the present study demonstrated that BGN is a prospective molecular target in BCSCs. Depletion of BGN in luminal cancer cell lines suppressed the enrichment of CSCs and tumorsphere formation. Furthermore, inhibition of BGN in BCSCs, CD29^hi^CD61^+^, ALDH^+^ and CD24^−^/CD44^+^ cells showed reduced migration, invasion and cell metabolism. The NFκB signaling pathway was attenuated with BGN deletion, suggesting that BGN regulates this pathway in BCSCs. Finally, BCSC mediated metastasis was decreased with loss of BGN, implicating a role for BGN in metastasis. Thus, BGN could be an important therapeutic target in breast cancer patients for the treatment of breast cancer metastasis.

## Figures and Tables

**Figure 1 cancers-14-00455-f001:**
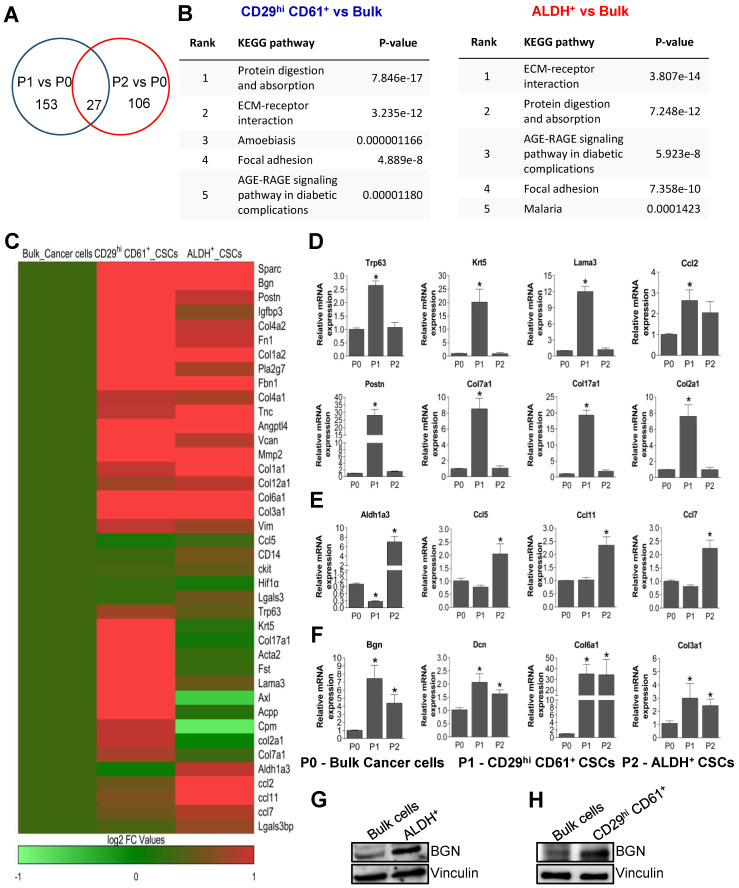
Upregulation of BGN in BCSCs derived from PyMT tumor cells. (**A**) Venn diagram depicting the upregulated genes in CD29^hi^ CD61^+^ (P1) and ALDH^+^ (P2) BCSCs as well as overlapping genes in both BCSCs as compared to bulk cells (P0). (**B**) KEGG pathway analysis showed that ECM receptor interaction and focal adhesion pathways were commonly upregulated in both BCSCs. (**C**) Heat map data showing the top 40 candidates identified from RNA sequencing. qRT-PCR data showing validation of genes in BCSCs, CD29^hi^ CD61^+^ (P1) and ALDH^+^ (P2) as compared to bulk cells, (**D**) expression of ECM genes, (**E**) BCSCs marker expression, (**F**) ECM genes upregulated in BCSCs. Immunoblots depicting expression of BGN in sorted BCSCs from MMTV-PyMT tumor cells (FF99), (**G**) ALDH^+^ and (**H**) CD29^hi^ CD61^+^ compared to bulk cells. Data shown as mean ± SEM of three independent experiments. Statistical significance was determined using two-tailed *t*-test; * *p* ≤ 0.05 as compared to bulk cells.

**Figure 2 cancers-14-00455-f002:**
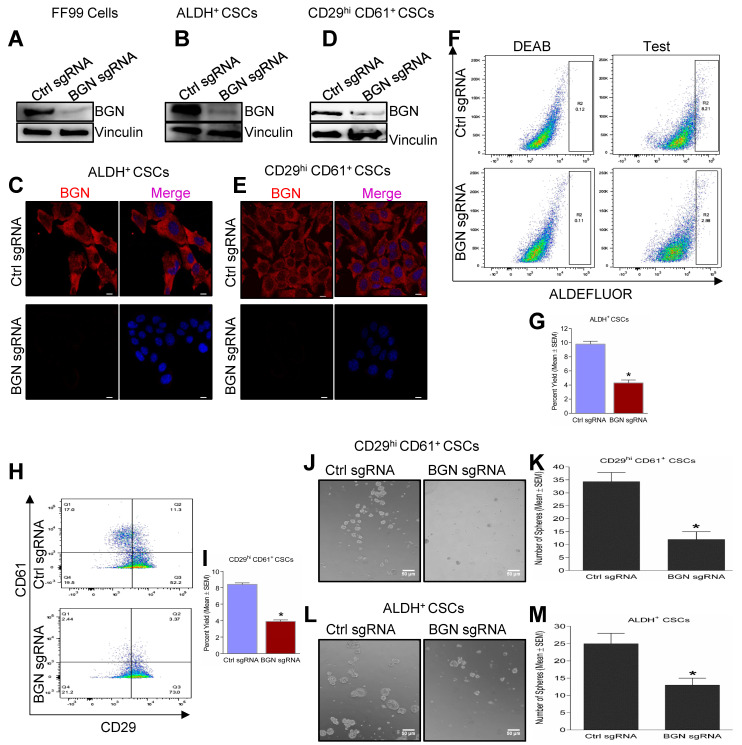
Loss of BGN in BCSCs leads to reduced CSC populations and tumorsphere forming ability. (**A**) Immunoblots showing BGN expression in MMTV-PyMT tumor cells (FF99) were stably expressing mouse BGN sgRNA and Control sgRNA (Ctrl sgRNA). Immunoblots and immunofluorescence depicting BGN expression in (**B**,**C**) ALDH^+^ BCSCs and (**D**,**E**) CD29^hi^ CD61^+^ BCSCs. (**F**) Dot plots of ALDH assay of ALDH^+^ cells in BGN sgRNA cells compared to Ctrl sgRNA. (**G**) Quantification of ALDH assay of ALDH^+^ BCSCs with BGN depletion. (**H**) FACS analysis of CD29 and CD61 marker expression in FF99 cells transduced with control and BGN sgRNA. (**I**) Quantification of H depicting the percentage of CD29^hi^ CD61^+^ BCSCs of BGN depleted cells. Representative images showing tumorsphere formation of (**J**) CD29^hi^ CD61^+^ and (**L**) ALDH^+^ BCSCs and their (**K**,**M**) quantification. Scale bar = 50 µm. Data shown as mean ± SEM of three or four independent experiments. Statistical significance was determined using two-tailed *t* test; * *p* ≤ 0.05 as compared to control sgRNA. Representative images of number of spheres were quantified using ImageJ.

**Figure 3 cancers-14-00455-f003:**
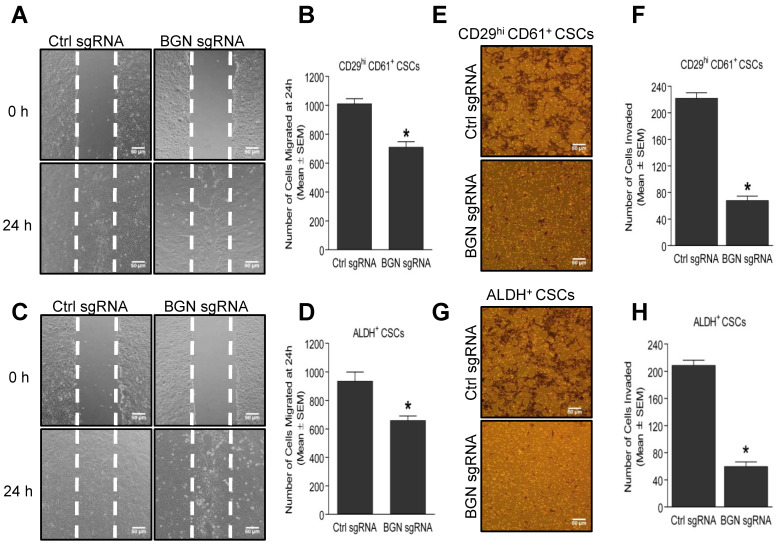
Loss of BGN reduces migration and invasion of BCSCs. (**A**) Representative images of wound healing assay showing migration of CD29^hi^ CD61^+^ CSCs with BGN sgRNA compared to control sgRNA at 24 h, quantified in (**B**). (**C**) Representative images of wound healing assay showing migration of ALDH^+^ BCSCs with BGN sgRNA compared to control sgRNA at 24 h, quantified in (**D**). (**E**) Representative micrographs from a matrigel invasion assay depicting the number of invaded cells in CD29^hi^ CD61^+^ BCSCs with BGN sgRNA compared to control sgRNA, quantified in (**F**). (**G**) Representative micrographs from invasion assay depicting the number of invaded cells in BGN sgRNA ALDH^+^ BCSCs, quantified in (**H**). Scale bar = 50 µm. Data shown as mean ± SEM of three independent experiments, *p* ≤ 0.05 (*t*-test) as compared to * control sgRNA group. Representative images of migration and invasion were quantified using ImageJ.

**Figure 4 cancers-14-00455-f004:**
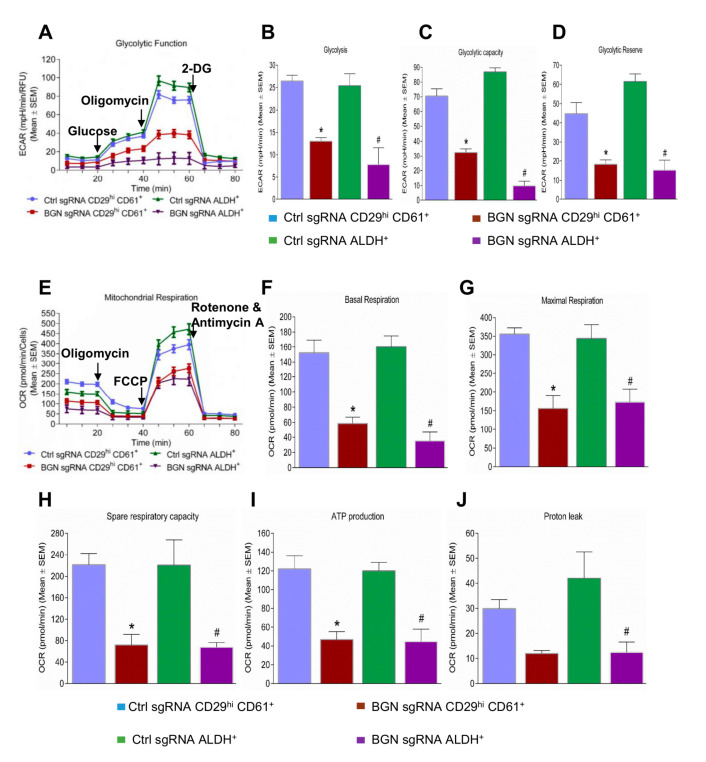
Role of BGN in metabolism of BCSCs. (**A**) A representative plot of ECAR from the XF96 extracellular flux analyzer of control and BGN sgRNA BCSCs in response to glucose, oligomycin and 2-DG. Bar charts showing levels of (**B**) glycolysis, (**C**) glycolytic capacity, and (**D**) glycolytic reserve in control and BGN sgRNA BCSC© (**E**) Representative plot of OCR from the mitochondrial stress test of control and BGN sgRNA BCSCs in response to oligomycin, FCCP and rotenone and antimycin A. Bar charts showing levels of (**F**) basal respiration, (**G**) maximal respiration, (**H**) spare respiratory capacity, (**I**) ATP production and (**J**) proton leak in control sgRNA and BGN sgRNA of BCSCs. Data shown as mean ± SEM of three independent experiments, statistical significance measured using one-way ANOVA (Tukey’s multiple comparison test) *****
*p* ≤ 0.05 of CD29^hi^ CD61^+^ and **#**
*p* ≤ 0.05 of ALDH^+^ as compared to control sgRNA group.

**Figure 5 cancers-14-00455-f005:**
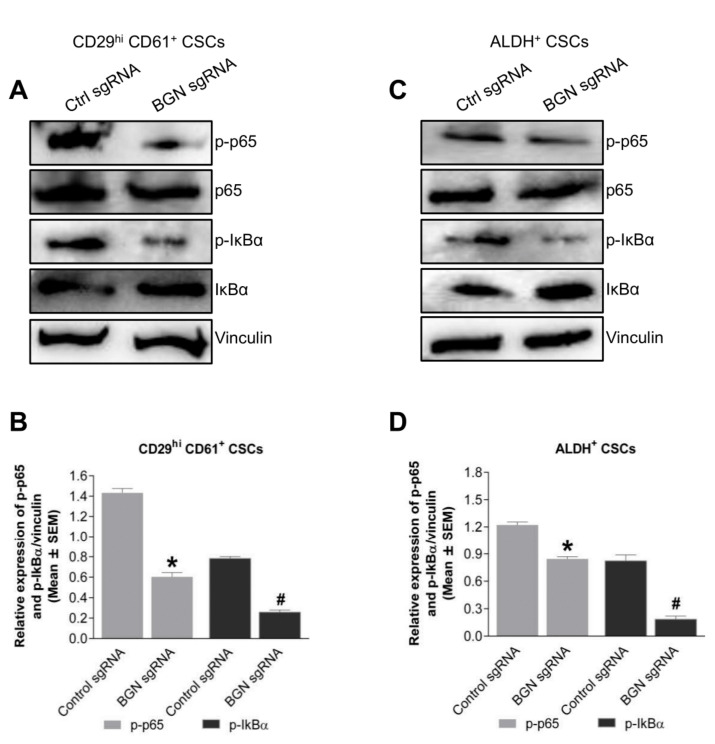
NFκB pathway regulated by BGN in BCSCs. (**A**) Immunoblots showing p-P65, p65, p-IκBα, IκBα and vinculin in control sgRNA and BGN sgRNA CD29^hi^ CD61^+^ BCSCs. (**B**) Quantification of A. (**C**) Immunoblots depicting p-P65, p65, p-IκBα, IκBα and vinculin in control and BGN sgRNA ALDH^+^ BCSCs. (**D**) Quantification of C. Data shown as mean ± SEM of three independent experiments, statistical significance measured using Two-way ANOVA (Sidak’s multiple comparison test) *****
*p* ≤ 0.05 of CD29^hi^ CD61^+^ and **#**
*p* ≤ 0.05 of ALDH^+^ as compared to control sgRNA group.

**Figure 6 cancers-14-00455-f006:**
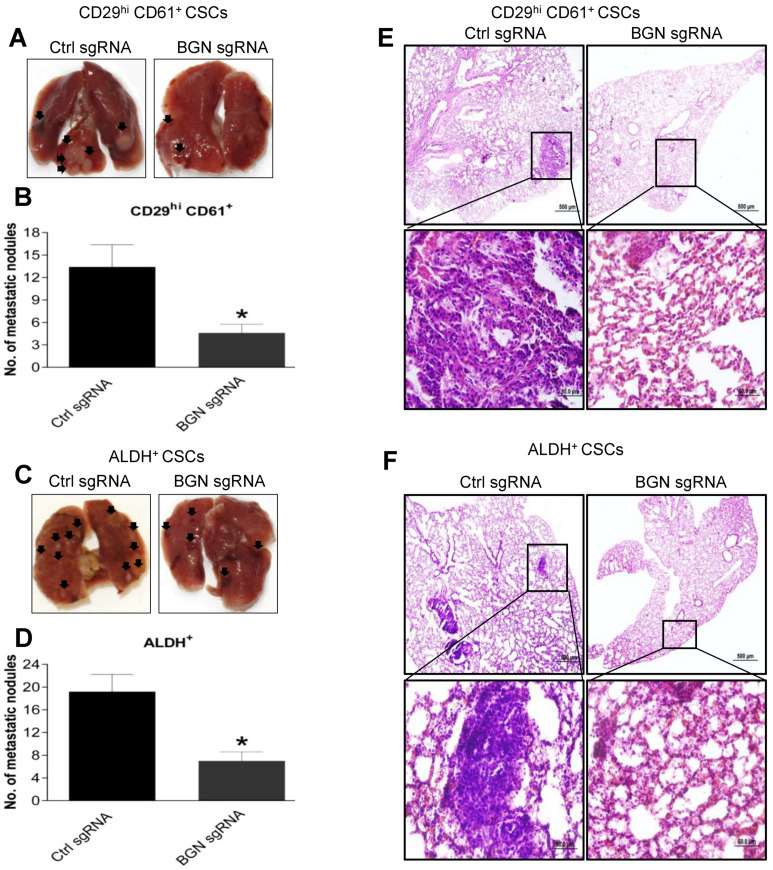
BGN promotes BCSC mediated breast cancer metastasis. (**A**) A representative image showing metastatic nodules on lungs from mice tail vein injected with WT and BGN depleted CD29^hi^ CD61^+^ BCSCs. (**B**) Quantification of A. ***** denotes *p* ≤ 0.05. (**C**) A representative image showing metastatic nodules formation on lungs from mice tail vein injected with WT and BGN depleted ALDH^+^ BCSCs. (**D**) Quantification of C. ***** denotes *p* ≤ 0.05. (**E**) H&E staining of lung tissues displaying metastatic foci in WT and BGN depleted CD29^hi^ CD61^+^ BCSCs. (**F**) H&E staining of lung tissues displaying metastatic foci in WT and BGN depleted ALDH^+^ BCSCs. Data shown as mean ± SEM. *n* = 5 mice per group.

**Figure 7 cancers-14-00455-f007:**
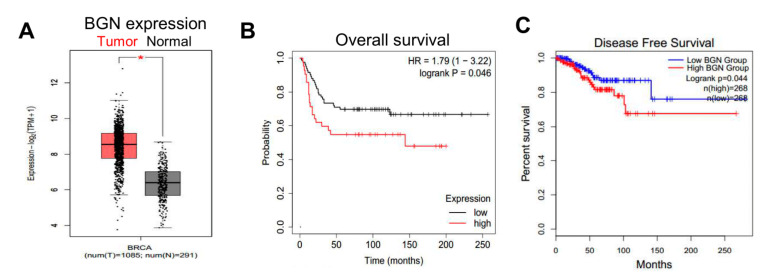
High BGN expression in human breast cancer tissues correlates with reduced survival. (**A**) Representative graph showing mRNA expression of BGN in human breast cancer and normal breast tissues. ***** denotes *p* ≤ 0.05. (**B**) Clinical data retrieved using KM plotter showing overall survival of breast cancer patients with high and low *BGN* expression. (**C**) Survival graph showing disease free survival of breast cancer patients with high and low *BGN* expression.

## Data Availability

RNA sequencing data were deposited in the GEO database under the accession number GSE189223.

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
