# Peer review of "Biglycan Promotes Cancer Stem Cell Properties, NFκB Signaling and Metastatic Potential in Breast Cancer Cells"

_cancers, 2022, doi:10.3390/cancers14020455_

Round 1
Reviewer 1 Report
Treatment of metastatic breast cancer is impeded by the persistence of cancer stem cells. A long-standing postulation in the field has been that eliminating these cancer stem cells could lead to cancer eradication and cures. Yet, the distinct and therapeutically-targetable markers of cancer stem cells have proven elusive. In this manuscript, Manupati et al identify biglycan (BGN) as a potential therapeutic target in breast cancer stem cells.
Overall this is a compact study examining the role of BGN in the breast cancer stem cells. Mechanistic depth is lacking, but this could be considered offset by the novelty aspects of the manuscript (i.e. the role of BGN in cancer stem cells has not been examined before). The validation of some findings in another model (MCF7) is also positive.
A weakness of the manuscript is the omission of experiments looking at the effect of BGN KO in non-CSC subpopulation, with regard to phenotypic endpoints such as migration, metabolic state, NFkB pathway or mammosphere formation. In the case of mammospheres, perhaps the mammosphere number and sizes would be much lower in e.g. the ALDH- cells, but in general this control condition would demonstrate that the effect of targeting BGN is CSC specific – a key claim of the paper. For instance, targeting BGN has been shown to have phenotypic effects even in bulk cancer cell populations (as described in reference #16).
Also, I notice that some mammospheres are formed even in the BGN KO conditions. Are these mammospheres generated by cells with inefficient BGN KO or are these stem cells bypassing the requirement for BGN to effect growth? This clarification would be important for any future development of therapeutic approaches targeting BGN in cancer stem cells.
Additionally, it is not very clear whether BGN KO causes apoptosis/cell death of cancer stem cells or just their proliferative arrest – this is also important for any translational ramification.
Finally, a proof reading is needed to correct some typos (e.g. the title of Fig 6 and other errors).
Reviewer 2 Report
The manuscript by Manupati et al. reports that Biglycan (BGN), a member of the small leucine-rich proteoglycan (SLRP) family, displayed tumorigenic behaviors such as BCSCs migration, invasion, and tumorsphere formation. The authors also demonstrated that BGN also plays a major role in the metabolism of BCSCs. Furthermore, reduction of BGN in BCSCs inhibits the NFκB pathway by reducing phosphorylation of p65 and IκB subunits. Overall the findings indicate that BGN could be used as a potential therapeutic target in BCSCs to prevent breast cancer metastasis. This is an excellent insight that will pique the readers' interest. However, a few suggestions will increase the quality of the manuscript.
1. Author should also show the role of BGN in chemoresistance.
2. There are a couple of grammatical and spelling mistakes. Minor changes to the manuscript's writing will also help to reinforce the paper.
Round 2
Reviewer 1 Report
The new data and modifications of the revised manuscript are satisfactory. The paper provides evidence for a potential role of BGN in the biology of breast cancer stem cells and will is of value for the field.